# Development of E-Service Provision System Architecture Based on IoT and WSNs for Monitoring and Management of Freight Intermodal Transportation

**DOI:** 10.3390/s23052831

**Published:** 2023-03-05

**Authors:** Dalė Dzemydienė, Aurelija Burinskienė, Kristina Čižiūnienė, Arūnas Miliauskas

**Affiliations:** 1Department of Business Technologies and Entrepreneurship, Faculty of Business Management, Vilnius Gediminas Technical University, Saulėtekio Av. 11, LT-10223 Vilnius, Lithuania; 2Institute of Regional Development, Šiauliai Academy, Vilnius University, Vytauto Str. 84, LT-76352 Šiauliai, Lithuania; 3Department of Transportation Logistics, Faculty of Transport Engineering, Gediminas Technical University, Plytinės Str. 27, LT-10105 Vilnius, Lithuania

**Keywords:** Information Communication Technology (ICT), Internet of Things (IoT), e-service provision system for transport monitoring (e-STMS), wireless sensor networks (WSNs), management of intermodal transportation

## Abstract

The problems of developing intelligent service provision systems face difficulties in the representation of dynamic aspects of cargo transportation processes and integration of different and heterogeneous ICT components to support the systems’ necessary functionality. This research aims to develop the architecture of the e-service provision system that can help in traffic management, coordination of works at trans-shipment terminals, and provide intellectual service support during intermodal transportation cycles. The objectives concern the secure application of the Internet of Things (IoT) technology and wireless sensor networks (WSNs) to monitor transport objects and context data recognition. The means for safety recognition of moving objects by integrating them with the infrastructure of IoT and WSNs are proposed. The architecture of the construction of the e-service provision system is proposed. The algorithms of identification, authentication, and safety connection of moving objects into an IoT platform are developed. The solution of application of blockchain mechanisms for the identification of stages of identification of moving objects is described by analysing ground transport. The methodology combines a multi-layered analysis of intermodal transportation with extensional mechanisms of identification of objects and methods of synchronization of interactions between various components. Adaptable e-service provision system architecture properties are validated during the experiments with NetSIM network modelling laboratory equipment and show their usability.

## 1. Introduction

Intermodal transport is used with different types of intermodal vehicles, and such kind of transportation has potential involvement in some risky situations. The problems arise when the planned processes are concerned with delays or unforeseen events in one mode and provoke disturbances in other transport processes. The unexpected disruptions in logistics processes can lead to the risks of not arriving on time and not getting to the right place. Aiming to avoid such problems of unsuspected events and inconsistencies in the consolidation processes, it is necessary to develop an e-service provision system that will be able to provide operative decision support for the management of real intermodal transportation processes. In order to provide solutions for the operational management of intermodal transportation situations, the aim of our research is to develop the architecture of an e-service provision system that would help to control the processes of cargo transportation in real situations. The requirements of the functionality of the system include the possibility of predicting some kinds of possible adverse situations and influencing the control changes in operational management through the proper support of e-services.

One of the problems of the development of the architecture of such type of intelligence system is related to the choosing of the infrastructure that will enable the monitoring of different kinds of intermodal transport components and will provide the control of the e-service provision for drivers, terminal operators, managers, and other decision-makers. The results of developing the architecture of the e-service provision system are related to our previously obtained results, where the complexity of intermodal transport is described [1,2], and the types of useful WSNs are classified. The components of ICT, which are applicable in the architecture of the e-service provision system, are classified by a large spectrum of functional abilities, including the set of outside equipment (as units) of transportation surroundings (like roadside units—RSU, with integrated different sensors, different communication channels, etc.), and on-board units (OBU), which can be integrated into vehicles. The possibilities of implementing WSNs and IoT are analysed with the integration of heterogenic infrastructures of communication equipment. Monitoring of dynamic objects and recognition of situations are related to managing the context of processes and are influenced by integrated components of operative and online management decisions [3,4].

Data obtained from different equipment using WSNs are stored in distributed stationers—host stations or servers working on the base of cloud computing technology. To ensure interoperability of data warehouses DWs and enable recognition of situations, the system of proper rules of big data warehouses is required with more detailed data mining methods [5,6], and these problems are outside of this consideration.

The strategy for digitalization of the European Union (EU) activates some critical interventions for adopting concrete Digitalization Actions Plans for ICT development in member States until 2030 [7,8]. The plans are related to the development of new integration of heterogeneous platforms, including the technology of IoT, the method for analysis of big data warehouses, the means of supporting integrity and interoperability of information systems, and communication platforms with WSNs. The efforts to have the whole service delivery process that takes place during the online regime and cross-borders are taken into account. The efforts of implementation of free movement of data, personal data protection, and cyber security are recognized in [9]. The General Data Protection Regulation and the Open Data Directives create strict obligations for EU countries [10,11,12].

New means are required for developing cybersecurity, which is mentioned by the objectives of the Information System’s Directive [13,14]. More vital attention to the cyber security objective imposes the obligations on the EU Member States to reach an appropriate level of preparedness against cyber-attacks.

Nowadays, transportation is closely related to intensive production, timely delivery to the market, and efficient supply chain management [15] with the help of ICT. When choosing intermodal transport services, increasingly more customers are concerned about the service quality, including cost and delivery time, avoiding various risks, and improving work indicators [16].

Therefore, increasing global competition encourages the efficient design of transport networks to ensure that they meet concrete sustainable environmental requirements. Since road transport has the highest CO_2_ emissions, intermodal transport could be a potential alternative. In this respect, intermodal terminals and their locations are critical in ensuring a sustainable transport network [17,18].

The functioning of an intermodal transport system depends largely on the locations and types/capacities of its intermodal terminals. Terminals ensure trans-shipment between the different modes of transport and are very important in organizing this process [19]. In addition to the primary operations, the following services can be provided in intermodal terminals: Temporary storage, if necessary, an inspection of transported goods, etc. [20].

Intermodal terminals are inevitably required to provide door-to-door services to customers, which combine collection and delivery services through the interaction of road and rail transport. This system benefits from good flexibility, mobility, and road accessibility while benefiting from the high cost-efficiency and environmental sustainability of railways, which ensures economies of scale [21]. It is, therefore, a practical option for replacing unimodal road transport of poor efficiency, which still dominates the freight industry in different continents [22,23], and is a viable solution for creating comprehensive and green logistics systems that support globalization and international trade possibilities [24].

Ensuring the interaction of various modes of transport, the smart service provision system allows for identifying the synchronization of the processes in usage and management of different modes of transport. Synchronization between transport means in reloading terminals is the basis for the right management of location positions during the time, thus transport infrastructure, modes of transport, and technologies in different stages [19].

The related work topics include the problems of the location of terminals, the selection of routes, network design, transport policy, and integrated location-routing issues [17]. Route optimization is widely recognized as playing a key role in optimizing the performance of an intermodal transport system [25], and it has been emphasized as a cornerstone of research in planning the transport network [26,27].

The interrelated processes of highway road–rail and/or rail–sea–highway types are analysed. Intermodal transport involves pickup and delivery, each of which separately affects the level of satisfaction of consignors and consignees [28,29]. However, the changing and uncertain environment of intermodal transport (road and rail) must pose risks and guaranteeing that a pre-planned route will successfully execute all transport orders is complex in an uncertain environment.

Therefore, routing capabilities are at some kind of risk. Route optimality is also at risk because there is no guarantee that the planned route will remain valid in a changing and unstable environment. Therefore, an essential part of transport planning is to minimize the risk [30].

The offered route should be analysed based on a realistic transport scenario. In a transport system, rail services have to follow the set values of timetables interrelated with intermodal transport means, for example, by the railway administration, which cannot be changed according to a route, whereas road services are flexible in this respect [28]. However, it should be borne in mind that traffic restrictions are usually applied in road transport, which is introduced to reduce congestion, pollution, and the number of accidents, and are especially strict when transporting hazardous substances [31]. Under such restrictions, the trucks involved in the traffic must follow the speed limits, which directly reduces transportation efficiency.

The importance of interoperability between seaport terminals and land-based intermodal terminals along international transport corridors is emphasized in [32]. The connections of intermodal land transport with seaports in international corridors are analyzed.

The aim of this research is to develop the architecture of the e-service provision system (e-STMS). We are formulating the requirements of such a service provision system that will enable the extension of the functionality of including ICT types, which will support the platform based on the infrastructure of wireless sensor networks (WSNs) and components of IoT. The recognition of obtained monitoring information has been applied to managers of transportation processes, drivers, and other participating persons in intermodal transportation. E-services are provided for implementation in freight transport and during the reloading processes in intermodal terminals. The approach is based on constructional methods for the development of the componential architecture of the e-STMS. The results obtained earlier by authors for describing the ICT infrastructure of intermodal transport management are also important here [4,5,7,8]. The research aims to analyze the possibilities of integrating heterogeneous services with ICT platforms that implement the functionality of a geographical information system (GIS), a geographic positioning system (GPS), and wireless communication channels. The e-STMS have created the need for online control and management procedures by recognizing the concrete situations of moving transport objects. The problems arise in the freight flow coordination in intermodal integration processes by focusing on safer and sustainable transport coordination and practical work of reloading terminals.

The paper consists of several sections: Section 2 is dedicated to the literature review, including the analysis of intelligent ICT usage in reloading terminals, analytical aspects of infrastructure construction for intermodal transporting of freights, and assessment of analysing impact factors. The proposed methodology that can be useful for designing the architecture of e-STMS is presented in Section 3, where we explore the possibilities of integrating moving objects into the infrastructure of IoT and implementing the WSN’s technological platform for monitoring. The structure of the components of the e-STMS is devoted to location identification and tracking intermodal transport objects and integrating them with the smart container management components (Section 4). The stages of workflow management in reloading terminals by implementing IoT technology are described in Section 5. The structures of transport service provision processes and issues of testing the system architecture are described in Section 6. The possibilities of experimental research by applying the NetSIM Lab equipment are demonstrated in Section 7. The analysis of the impact of the proposed architecture, limitations, and plans are presented in the Discussion part. The summary and scientific novelty are shown in the Conclusions.

## 2. Related Works

The technology of IoT can include physical devices like remote sensors that can collect and transmit large amounts of data and significantly affect the operations of endpoints [3,4,6]. The IoT technology is integrated into the whole system infrastructure and is helpful in operational planning and management activities, i.e., inventory delivery, tracking, and forecasting, decision-making in terminal operations.

Integrating technology platforms allows businesses to track products throughout their lifecycle using RFID and GPS sensors. Intermodal terminal managers can collect valuable information about durations of product storage, temperature conditions of freights, the delivery time of each transaction, and the shipping time spent in the terminal. With different sensors, it is possible to monitor various conditions. The systems can display inventory levels in real-time moments, and managers can analyse other trends and predict plans for future operations more accurately. Installation functions can use data from IoT and display data for better determination of critical metrics that increase the reliability and efficiency of their services. However, all these possibilities are enabled if the work of components is integrated into the IoT platform safely and correctly.

Delays in terminal operations can help managers to identify the root causes of bottlenecks and take steps to improve the process. Since the location of goods can be tracked in real-time during delivery, the dispatch centres can provide customers with excellent customer service by delivering accurate delivery plans and real-time location information.

Expanding inland terminals helps seaports alleviate the inevitable problems of storage capacity. Seaports are appealing because they are reliable and connect remote points [33]. Therefore, the processing of container storage is one type of operations which are important in all terminals. Optimal storage refers to proper management, fast retrieval times, and minimal unproductive steps, defined as factors that keep the container from getting closer to its final destination (for example, reaching a specific container, among many others) [34]. In light of the above, the optimization of container distribution and collection methods can be said to depend on the availability of storage space, the degree of mechanization, and the management policy [35]. Whatever the storage facility, the distribution methods used in inland terminals should coordinate technological processes in seaports to ensure optimal handling of containers.

Usually, research on seaports focuses on increasing the efficiency of quays to speed up container ships’ trans-shipment [36]. The analysis of intermodal transport mainly focuses on operations in intermodal terminals, the design of this transport network, optimization of intermodal transport routes, and synchronization of processes [37,38,39].

As the primary node of the intermodal transport system, intermodal terminals provide the necessary equipment and space for the trans-shipment of containers between different modes of transport (sea, rail, and road transport). The majority of research on intermodal terminal operations focuses on ports of the container. Some activities of container ports are examined from different perspectives by different authors (Table 1).

However, researchers have mainly focused on container ports, paying less attention to optimizing rail and road intermodal terminal operations. Although rail and road terminals and container ports have similar handling equipment, the rules and operating procedures of these two types of intermodal terminals are very different. When comparing operations at container ports and road-rail intermodal terminals, the critical difference is that the area of ship loading in a port of container and the yard of the container are located in the same loading area at rail-road intermodal terminals.

In intermodal terminals (railroad), gantry cranes are mounted on rails and responsible for both loading-unloading and storage operations at a time, while in container ports, this is done by quayside cranes and local cranes separately [55].

Various processes in intermodal container terminals daily are often analysed in isolation, not considering their overall complexity [56,57,58,59,60,61,62,63,64,65,66,67,68,69,70,71,72,73,74,75,76,77,78,79,80]. Some works provide methods for the optimization of operations of container ports. The differences in operating procedures and rules between the two types of intermodal terminals make applying the technologies for existing rail and road intermodal terminals difficult.

## 3. Methodology of Construction of Architecture of e-STMS Based on IoT and WSNs

The methodology for designing the architecture of working online the e-STMS for cargo intermodal transportation monitoring and management is presented by integrating different layers of the whole infrastructure. The requirements for such system architecture are formulated by taking into account the conditions of transportation, which can vary in a broad spectrum. The conceptual models are combined with two main subsystems: The functional subsystem, which is devoted to the representation of interrelated processes of intermodal transportation, and the e-service provision subsystem (Figure 1). The flow of data (information) between functional subsystems and components supporting such subsystems must be accessible through separate channels and enable the transmission to each participant component by taking into account several steps of realization of the whole infrastructure (Figure 1).

For creating service areas for everyone involved in the processes (i.e., customers, carriers, small charging stations, charging partners, and contract drivers), it is possible to monitor the logistics and intermodal transport processes in real-time and increase the company’s transparency. The development of new projects and decision-making processes depends on the revenue stream, affecting the efficiency and effectiveness of all consolidation works (operations) in the intermodal transportation of freight. The system can schedule data, including WSNs and other equipment, by providing data to operatively working big data warehouses in stationary working servers. Processes are organized so that interoperability of such infrastructure enables data collection from sensors automatically. Each step in the process requires standardized decision-making on appropriate planning, urgent anticipation, joint implementation, monitoring, and service quality. Factors that affect the final result have to be taken into account. This analysis is carried out to assess customer satisfaction with the e-service provision for business partners’ activities during Step 7. In more detail, we are discussing the means for realizing Step 2.

The types of vehicles and interaction scenarios are analysed by considering the complexity of intermodal transportation. Vehicles can move from low to very high speeds (160 km/h or more), making it difficult to maintain consistency in coherent infrastructure-to-vehicle (I2V) and vehicle-to-vehicle (V2V) communication [3,8]. The existing statistical data on vehicle traffic, such as the movement together according to specific patterns during peak periods and under different conditions, could help to keep a link between the mobile automotive groups. Vehicles at any time may be out of communication coverage (in some cases of conditions the techniques based on Wi-Fi, cellular, and satellite communication does not work), so the network protocols have to be designed in such a manner that they can easily connect to the Internet and provide support as in normal mode. Despite the many positive, unique features, vehicular network development is faced with specific challenges, such as:Large-scale different connections into networks;High level of mobility of observed objects;Fragmentation of the network;Changing topology of freight transportation;Complex communication, accounting, and quality assurance.

In principle, vehicular, and environment communication networks could extend across the road network and cover many types of network equipment (i.e., vehicles, intermodal terminal objects, like cranes, cargo containers, warehouses, etc.).

The layers applicable for monitoring moving objects involved in intermodal transportation are built and based on a secure working IoT infrastructure. Such an infrastructure is analysed according to the possibility of dividing the layers of application, network, and perceiving (Table 2). A deeper structure of procedures is used to identify the concrete moving objects. Such structure is described in the next section by representing the optional, required operations at some layers by representing the algorithms that are necessary to empower the IoT platform.

More in detail, the data accessing and identification problems of moving objects are analyzed, and all these issues of development of layers must be solved for the safety integration of IoT and WSNs into the e-service provision system (e-STMS).

The security requirements have to be formulated to develop the e-STMS by enabling the integration of moving objects in the IoT technological platform. To ensure the possibility of recognition of needful kinds of intermodal transport objects, the security requirements are related with:data authentication;access control;client privacy;attack resiliency;secure bootstrapping and transmission of data.

The implementation of IoT has some limitations, like energy restraints and low computing power. For organizing the processes of data transmission about real situations of moving objects (i.e., transport means, trailers, freights) from different kinds of Road Side Units (RSUs) and On Board Units (OBUs), we have organized the recording and accumulating of such data into the distributed data warehouses, which are working on the platform of cloud computing of stationary working servers. There, the system implements all necessary communication protocols. On the layer of data transmission, all layers’ interoperability possibilities have to be supported. Our consideration is limited to the analysis of identification problems that arise when determining the activities of transport and other objects involved in consolidation processes, which are needed for the development of the e-STMS (Figure 2).

The intermodal transportation environment networks operate in extremely dynamic conditions, which can sometimes be highly different. For example, on highways, speeds could reach up to 130 km/h, in low-density roads, car density may be only about 1–2 cars per km. On the other hand, the speed of vehicles in urban areas is 50–60 km/h, and the transport flow density is relatively high, particularly during peak periods. Often, vehicular communication networks could be fragmented.

The dynamic nature of traffics can lead to significant gaps between cars in sparsely populated areas. It can also be created a few isolated clusters of network nodes. Vehicular communication networks’ scenarios highly differ from the classic ad-hoc networks since the cars are moving and constantly changing positions, and scenarios are highly dynamic.

Furthermore, the network topology changes frequently due to frequent connections and disconnects between network nodes. The degree to which the network is combined depends on the distance between the wireless nodes and the number of connected vehicles [5].

An application device like OBU covers such main functions: Wi-Fi access, Ad-Hoc issues solving, geo-location-based routing, network load management, reliable messaging, security, and IP mobility [81].

Some system components are connected into subsystems for cargo transportation management and scheduling. The e-STMS is divided into some packages for service provision for transportation managers and drivers (Figure 2).

The set of RSUs can be facilitated by WAVE devices installed near the road or in other specially designated areas, e.g., at intersections or parking lots. Such devices are equipped with short-range wireless technologies, such as IEEE 802.11p or others that connect them to the network (IoT) infrastructure. If direct communication is not possible, the data are transferred to other cars as intermediaries until the addressee is reached. These forms of a multi-hop relationship are described in [81].

The infrastructure of the domain is related to all these parts, especially for the RSUs connecting infrastructure of networks or by the Internet by providing OBU access to that network. OBU devices communicate with various nodes by providing non-security services using other mobile technologies (GPRS, GSM, 4G, 5G, HSDPA, UMTS, and Wi-Max) [82,83].

To improve the quality of e-services, it is necessary to manage the flow of information between companies organizing transport between trucks, customers, and logistics representatives from different international centers, as well as forwarding companies to international delivery terminals through or to customers from other countries.

At the same time, the effectiveness of this model is closely related to the transport management subsystems, which are based on the infrastructure of work of operators by accepting the orders and using various means, such as e-recommendations, e- reports, by using different types of mobile applications for drivers and customers.

## 4. Application of the Subsystems for Management of Logistic Processes in Reloading Intermodal Transport Terminals Based on IoT

Logistics and freight transport are areas that currently have competitive advantages in applications of intelligent systems based on innovative ICT. These changes have led to the emergence of new paradigms for the design of logistics systems following various criteria—activity, consumption of energy, the performance impact on the environment, etc. [84,85,86]. For example, the European Super Green project [87] aims to help define and compare freight corridors across Europe from technical to economic and environmental to spatial investigations. This was reached through intelligent ICT methods for corridor assessment [88]. With the widespread integration of innovative ICT, new communication management models have become possible. The modern terminal is characterized by container traffic and integration into the logistics network, in which land, rail, and sea transport ways are integrated. The efficiency of the terminal is greatly affected by its ability to establish contacts with areas further on, allowing goods to reach their destination quickly. In addition, the EU promotes speeding, safety, and the rationalization of freight transport by other modes of transport through several initiatives.

The integration of ICTs started in the year 2000 with solutions, which included the emergence of web services (XML) and service-oriented architecture (SOA) [89]. These smart devices were connected using the convergence of many technologies having wired connections to the Internet, integrating systems using sensors, and communicating with IoT technology platforms [90].

The challenge of intelligent ICT systems integration arises because of their complexity and the need to adapt technologies for new logistics requirements [91]. To revise the impact of intelligent ICT usage in the communication of terminals, we have to figure out specific requirements and then find out how the application of ICT will fulfill these requirements [92]. ICT became a fundamental element [93,94], taking into account:logistics process is considered an integrated process used for value creation;reducing expenses of operations in the chain;communication between the companies;improvement of performance;ICT is also available at accepted quality level (or performance level) and transmission service (or interoperability service).

In principle, the projection and realization of integration could be treated as a task for applying various intelligent ICT-based systems working in a concrete sequence of management stages. The main objective of innovative ICT is to help manage terminals’ work by integrating tasks and resource allocation (humans and machines) in many repeated cases. It is not enough to provide employees with appropriate innovative ICT tools. However, with the provided infrastructure, they take into account such links between the different activities of different employees and improve their cooperation and execution of all stages of any workflow. Describing models could support conceptual solutions of describing conditions to perform tasks in an integrated way [95]. ICT is part of executive models, which are used for implementing process cases on supporting by ICT platforms [96]. It could effectively suggest alternatives to management or technical decisions in multiple cases. Various workflow models could be combined and captured, thus creating a coherent environment for integrating complex mutual behaviour [97].

An approach to assessing improvements is an advanced ICT application in logistics terminals. First, there is a literary review of intelligent ICT for terminals. Secondly, the assessment model used to evaluate the impact of both ICT-based solutions: RFID and WSN at terminals, will be suggested. The modeling results show that it is possible to save performance lead time and the use of resources significantly. The proposed approach depends on the terminal and the ICT solutions considered to be carried out for the feasibility study.

The implementation of intelligent ICT aims to solve various problems of terminal management. It is recognized that vertical solutions, which involve sensors and gears to control the terminal yard, have been used since the 2008. The paper [98] shows examples of port logistics that benefit from investments in intelligent shipyard management techniques and semi-automatic or automatic technologies managed remotely. Automatically connected controlled machines (AGVs) and automatic loading cranes (ALCs) can increase container movement speed. Automatically connected controlled machines are robotic vehicles that drive on a predetermined road identified by electrical wires inserted into the network. ALC moves along the rails and is managed by the centrally organized system. Following the minimal space requirement, ACLs can ensure the storage of high-density containers. For example, some terminals in the port of Rotterdam use both AGV and ALC techniques [99].

Sensors are also used to increase the safety of crucial terminal zones like gas and oil stations and military service [100]. Terminal protection includes various measures to control entry into the yard and the side of the entrance. This includes crews, passengers, cargo carriers, and drivers. The complex technique is the autonomous monitoring system that detects “abnormal” events for the quick reaction of a particular operator. This requires processes of extraction, recognition, and correlation of functions [101]. If intelligent ICT solutions cover the usage of various techniques in different areas of the terminal environment, we will find works on the optimizing of planning and organizing the operations of terminals [98,99,102,103], strengthening resources of transport systems [104,105,106], provision of real-time positioning systems [107,108]. Terminal management is also moving towards an advanced ICT-based integration infrastructure to shorten transport times and increase logistics efficiency, focusing on cloud systems [109,110]. This provides critical capabilities through ubiquitous computer technologies that enable real-time logistic synchronization.

Such functions could be grouped into these sub-groups:Retirement of locations, tracking and identification of objects, andFreight traffic management, e.g., carrying products, is tracked, seeking to reduce economic loss.

Innovative ICT solutions can improve the formation of terminal work by implementing innovative means, like:WSNs and IoT for localization, tracking, and identification of transportation objects;Using several intelligent methods in e-service provision systems for managing intermodal transportation of freights.

Such main applications are split into the following subdivisions, considering hardware, software, and related technologies.

## 5. Location Identification and Tracking of Intermodal Transport Objects and Integration with Smart Container Management

In location identification and ready GPS solutions, some of which support freight delivery to terminal scenarios, some additional issues need to be addressed. GPS-based solutions, for example, can monitor the state of transferring of cranes and containers by using the data from central databases and are related to an object moving positions of the concrete geographic coordinates at the fixed time moments. Such solutions perform well in principle, except for some limitations, as containers are sometimes transported by trucks and tractors. GPS solutions cannot always track yard trailers, as giant cranes and container chimneys cause many dead areas.

An alternative GPS solution system is the real-time positioning system (RTPS), which displays the concrete position of the container when the tag of such system is attached [111]. As a rule, locating the premises includes the small nodes and predominant method resulting in a Signal Strength Indicator (RSSI) [112]. Various solutions could be used in this area, ranging from Wi-Fi [113], Bluetooth [114], and WSN [115] to ultrasound [116], long-range radio frequency technology [117], and short-range radio technology [118].

The well-known RTLS technology in the port logistics scenario is identification following the Radio Frequency possibilities (RFID) [111,119]. RFID solution could indicate location, which is related to an object’s position and its unique number. Such properties allow for more efficient management of container terminals because containers are quickly placed but are less helpful in identifying their location. In most cases, RFID requires infrastructure to scan tags and involves human or semi-automated operations. With the help of up-to-date systems, they help replace the physical search of containers placed in the wrong area). Tracking and tracking capabilities of localization methods using wireless sensor networks [120,121]. WSN has proven itself in a variety of scenarios, from detecting human activity [120,121] to everyday life in the environment [122], by helping to identify spaces in the system [123,124]. The researchers suggest an advanced approach helping continuously to identify container position by using the wireless network of sensors [125]. Each container has multiple nodes that use wireless communication to discover ‘neighbours’ containers. Recently, the hybrid RFID-WSN method has been connecting the main aspects of both technologies, including a new platform allowing active tag placement to monitor product temperature [94]. Authors improve RFID tags with enhanced capabilities for communication [126] through the network providing a way to integrate RFID with WSN to extend typical RFID identification capabilities with localization capabilities enabled by WSN.

Considering the unique indicators presented in the literature, we select RFID and WSN-based RTLS in our model as reinforcing factors for locating, tracking, and finding containers according to the cases under consideration.

In the e-STMS, we use the IoT platform for object identification. The set of observed, moving intermodal transport means can be described as a set of objects *Observed Object Set = {object_i,j,k_}* in the system’s infrastructure. Going into the deeper layer of identification and authentication of the moving objects, we represent them as:

*Object_i,j,k_*, with some types of indexes:-Index *i* represents the real identifier of the transport object that is observing and monitoring in the system;-Index *j* represents the coordinates of the location of objects recognized by RSU;-Index *k* represents the time when the object is recognized in the *j* location.

The structure of assignation of *Object_i,j,k_* to the IoT of the objects for secure management can be designed as the multi-layered, i.e., layer for cycles of processing the object. Fog layer and procedures of object application in IoT infrastructure are considered (Figure 3).

The registration of moving transport objects is related to some security components of integration layers, presented in the previous section (Table 2). The Object *O_i,j,k_* identification and authentication methods for including an object in the IoT platform are realized in the Fog layer using blockchain technology functions. The identification and authentication process starts when the object initializes the data transmission process. The scanning of prompts of *O_i,j,k_* is realized by Sensor *S_l,m,ti_*, where *l* is the identification of Sensor, *m*-is identification of the geographical location of Sensor, and *t_i_* is the time moment when the Object *O_i,j,k_* is scanned. Data are transmitted to the Fog layer for realizing the stage of registration object *O_i,j,k_* implementing a more secure identification and authentication process. The message is transferred to Message Broker in the Fog layer, which forwards the message for processing.

Open-source systems and tools have been selected for the methods which allow the development of the prototype system for transferring data to the cloud or virtualization platforms. That data streams are used for further functionality development. The experimental possibilities of these functionalities are shown in Section 7 by applying NetSim network modeling Lab equipment. The architecture is modular, supports the possibility of extensibility of functions, and allows adding new specialized servers or containers to the Fog layer.

The information received during message processing from the objects is transmitted to the aggregation servers, systems, or applications to perform further operations with the received data. The representational state transfer (REST) application interface (API) server allows operations to be performed in a blockchain, so new system components can use defined API references to provide additional functionality.

Advances in machine-to-machine communication (M2M) solutions and WSNs are significant for terminals. They are used not only to find or track freight on its way during delivery. In addition, WSN helps to monitor freight delivery conditions during its journey. It also alarms if the container is opened with the help of sensors (e.g., magnetic contact sensors) or shows temperature fluctuations (with the use of temperature sensors), indicating all details about the occurrence of these events and reporting the data over the Internet. With the help of sensors placed in the containers (which become smart), emergency cases (drops, fires, floods, or others) could be reported by sending SMSs about the event to insurance agencies to call for emergency assistance. It could also be checked if working hours correspond with GPRS/3G alarm timing.

Three main features are enabled by WSN using in the infrastructure of smart containers:detecting unexpected opening of doors,monitoring freight delivery circumstances,identifying storage conditions by adding sensors measuring environmental variables for freight’s sensitivity to risky conditions [127,128].

In the case of fragile goods, detecting shock and vibration exposure 3-axis accelerometer sensors can help to identify the issues. Finally, sensors can actively use RFID and Near Field Communication (NFC) technologies to exchange information with other nearby stored pallets or containers. For example, the alarm could be generated when dangerous freight is placed next to flammable substances [129].

## 6. Workflow Management in Reloading Terminals by Implementing IoT Technology

The terminals are where containers arrive by international freight delivery modes and are handed over to domestic freight forwarders, such as land trucks, rail, and other ways around. The terminals have such main functions as reception, storage, recognizing aging, and reloading.

More specifically, the terminal should be equipped with coastal cranes for unloading containers from ships. Unloaded containers are transferred into yard premises (storing), usually divided into subzones for trade, exceptional, and empties. The freight can be delivered by land trucks, other transporters, and AGVs, helping to prepare them for departing or arriving from/to storage areas.

Since there is not enough information helping to determine the duration of storage (temporal and medium duration), sometimes it is necessary to perform unpacking and consolidation in the deconsolidation zone (where deliveries from different vendors are repacked) and consolidate there. If no consolidation is required, the container is automatically transferred directly to the staging zone, which is used for train or truck loading.

To measure the impact of RFID and WSN systems on an end-to-end workflow which could be used to speed up the tasks of inspecting the truck and determining the container’s location. In general, it could be stated that without the application of RFID, several steps of check-out operations have to be performed, such as stopping the truck, handing over copies of papers to the doorman, and looking for permission before restarting the land truck.

On the contrary, when using an RFID system, the activity is completely performed without manual intervention because RFID scanning allows you to set and control the container, enabling you to open the truck gate.

Sometimes additional documents are required, so the activity takes around 30 s. Such a method works perfectly at various entrances of terminals. If the use of WSN technology for the localization of containers is proposed [124,130], WSN detects containers from neighbours. Geometric boundary data are combined to revise the container’s position and thus accelerate its location.

The identification of existing localization is delivered with GPS. GPS allows you to track the location of crane-handling containers. Such a system is usually adequate, but it could have several restrictions because the crane does not entirely transport the container, but land trucks and other transporters are also used.

RFID solution quickly identifies containers but is less helpful in figuring out their location. In addition, RFID solution requires infrastructure to identify tags and the process, which in most cases is controlled by the person. With the current solution’s help, real-time tracking containers require human involvement.

Sensors are important elements for IoT functioning. The sets of data that are created by these sensors are large. Sensors communicate by using regular channels, such as cellular or Wi-Fi networks. The collective bandwidth application is available for data transfer at the sensors. The approaches that can help equip end node sensors with processing capability to select data for interpretation, analyse it, and provide status are vital (see Table 3). Information collected by sensors is transferred to the cloud for further revision.

## 7. Structure of Transport Service Provision Process

Conceptual modelling is carried out according to specific scenarios and gradually modifying the main parameters. The method is named “what-if scenario” analysis and covers data-oriented modelling activities when the purpose is to test the behaviour of part of the company’s model according to specific cases. Practically, “what-if” scenarios help to measure changes in the parameters, which affect the process (b). Shortening the process’s duration is possible by introducing the e-STM activities and helping to increase the productivity of the intermodal terminal’s process. We can describe some cases of the impact of two improvements that reduce consolidation resources.

Among the possible obstacles to implementation:
(i)The application of RFID sensors under truck gates led to the obtaining of multiple devices in different cases effectively.(ii)The application of WSNs is usually handled through a general standard solution of smart container in case of full load delivery via intermodal terminal. It is more convenient to install other sensors adopted to use different world standards to increase scope, strength, and interoperability.

Improving the quality of services during transportation will be hampered by different interpretations of the quality of road transport services, which should carry less cargo than cargo [137,138]. Customers ordering the delivery of vans using e-service provision system technology believe it is necessary to obtain all information about the order quickly, and business partners should use subsystems for e-service provision as well as interfaces for monitoring all types of needful observing objects. Measures are ensured for the quality of shared services. A conceptual model for improving the quality of e-services, in which all processes, including participants, are exposed into our system (e-STMS), is presented (Figure 4) and is oriented for employees (drivers) and customers of inter-consolidation processes (i.e., transport companies, reloading terminal operators, etc.). To do this, combining the maximum number of participants in the process into one system to improve its quality is necessary.

## 8. Illustration of Experimental Possibilities for Obtaining Results with IoT and WSNs Connectivity by Using NetSIM Laboratory Equipment

For the experimental research of IoT connectivity possibilities of experimenting with a wide range of wireless sensors, the NetSIM network modelling laboratory equipment was used [137]. The NetSIM provides the equipment of the enhanced Laboratory (Lab) with possibilities to configure the Enhanced Interior Gateway Routing Protocol (EIGRP) and apply the Message Digest 5 (MD5) authentication properties [138].

The NetSIM Lab enables routing table optimization using multi-area Open Shortest Path First (OSPF) by standards of CISCO. It is possible to explore the usage of a summary of routes and observe the effects on the size of routing tables. This Lab covers OSPF basic authentication as well as OSPF-encrypted authentication. To illustrate how it is possible to connect the configuration of routers and other workstations using NetSIM Lab, we present the example of configuration possibilities for experiments (Figure 5).

The Enhanced Interior Gateway Routing Protocol (EIGRP) experiments with the authentication process of objects used in the NetSIM Lab environment. It enables configuring the routers in the simulated network, which require EIGRP authentication before they advertise routes to or accept routing table updates from EIGRP neighbours.

For the simulation of multi-area components, it is possible to configure a multi-area Open Shortest Path First version 3 (OSPFv3) working on the base of the Internet Protocol version 6 (IPv6) network.

The Open Shortest Path First (OSPF) authentication process enables configuring the routers in the simulated network, which requires OSPF authentication before they advertise routes to or accept routing table updates from OSPF neighbours. This Lab covers OSPF basic authentication as well as OSPF-encrypted authentication.

The experiments were provided by testing the stack work varying the number of connections of different amounts of IoT devices (by providing investigations with 10, 100, 1000, 10,000, and 100,000 devices) with varying protocol options (Figure 5). The specification of concrete such protocol options is chosen from the systemic network layer. The results have shown the time consumption by uploading the different amounts of devices and illustrate time differences—how long they work under additional conditions of procedures of the identification and authentication stages. Obtained time durations according to the more significant amount of connectivity were proportional longer according to connections of the larger number of devices. The work, by adding some safety means, also influences longer stack work.

The RF (Radio Frequency) possibilities include propagation for the wireless link, the layer of wireless connections enabled by the standard Wireless Protocol layer.

During experimental research, data flows between devices are analysed by connecting different amounts of devices. Time consumption for data exchange and server work duration are evaluated according to different amounts of connected devices and algorithms applied to ensure security (in experiments with 10, 100, 1000, 10,000, and 100,000 devices).

Additionally, experiments with different identification algorithms can be performed and may include the encryption models, such as AES, DES, XOR, and TEA, which are supported for encrypting the application payload. The NetSim has inbuilt interfacing with “Wireshark” that can be used to capture packets from the virtual nodes during the simulation.

## 9. Discussions

The processes of intermodal freight transportation management are analysed to develop the system that helps monitor and control such participating components. The proposed e-STM structure is working on the base of equipped communication infrastructure of different communication channels, WSNs, and on the base of the implemented platform of IoT technology. For the development of the e-STMS—adaptable in-service provision for managers, drivers, and persons participating in consolidation processes we have constructed the object’s identification and authentication infrastructure based on the safest means of IoT. We have analysed all participation components of intermodal transportation processes related to the complexity of surrounding equipment, heterogeneity of communication channels, and complex managerial computer-based components.

An approach for integrating heterogenic sources of monitoring data during freight intermodal transportation processes is presented. We hope that the described means can help in the provision of e-services needed for the safety and online management of intermodal transportation. A review of related works helps understand how intermodal transportation processes are analysed and what data are important in different management stages. The presented approach intends to develop an e-service provision system adaptable for intermodal transportation with some extended properties of integrating IoT platform possibilities. The proposed structure for the identification and authentication of moving objects in IoT platforms helps the development of safer means to omit unsuspected activities from such objects. Our experimental studies show advantages in building an infrastructure of e-STM with heterogeneous components to integrate wireless communication channels. Our future research investigations will address the more complex scenarios for accident event recognition and provision of prediction of future unsuspected situations. In the future, the classification of infrastructure components according to various types of wireless technologies and the transfer of context-oriented data for specific smart service designs will be implemented to reduce the number of accidents and unsafe situations.

Illustration of experimental research using NetSIM Lab enables the evaluation of IoT connectivity properties and their work under different ranges of connected sensors. The construction of the extensional potential of the multi-layered intermodal transportation infrastructure will be the focus of future research with the inclusion of intellectual recognition possibilities and context-aware information implementation.

## 10. Conclusions

The described approach is based on the extensible infrastructure of the necessary components for developing an e-service provision system for safer monitoring and controlling moving objects participating in intermodal transport processes. Some components are integrated into the system development process by the inclusion of features and algorithms for extension functionality of infrastructure of IoT and WSNs. The proposed architecture of e-STMS helps monitor and control transport means more safety by including obtained data from sensors equipped using different communication channels. The proposed structure of identification and authentication of moving transport objects will help gather data from WSNs more safely and integrate them into the infrastructure of IoT.

The authors proposed the main steps of action for implementing the necessary processes for developing the e-STMS.

The proposed extensions of the structural architecture of the e-service provision system are related to our previous results, but these results are unique according to the created algorithm for identifying moving objects. The presented structure of the system enables the development of adaptable software means, which will be necessary to automate the management of intermodal transportation situations. To address the challenge of the proposed system, it can allow the usage of environmental context information, including vehicle dynamics and information transmission from vehicles to IoT infrastructure.

The study has some limitations according to the description of details of the authentication process that could be analysed in future studies. The typology of unplanned events is not presented.

The proposed e-service provision system will be helpful in analysing the heterogeneity of communication channels, and the service will be adaptable for more secure security, including moving objects into the IoT platform.

## Figures and Tables

**Figure 1 sensors-23-02831-f001:**
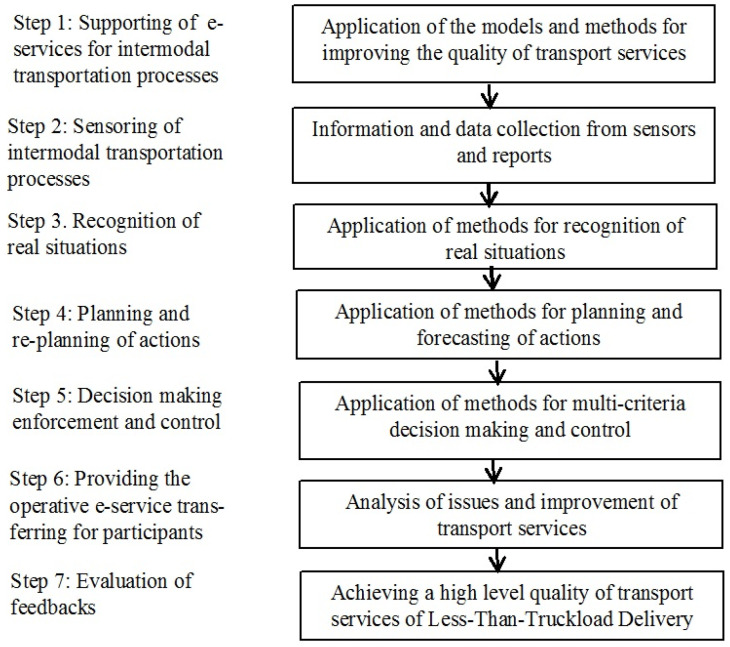
Main steps of realization of the infrastructure of the e-STMS.

**Figure 2 sensors-23-02831-f002:**
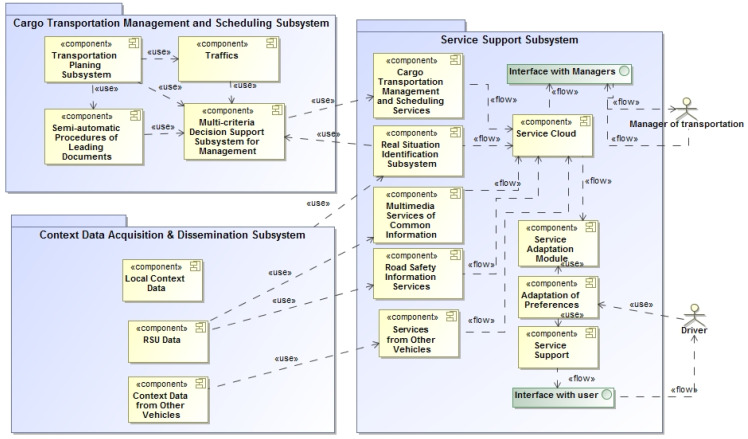
The componential view of e-STMS architecture for service provision to drivers and managers who are participating in intermodal transportation.

**Figure 3 sensors-23-02831-f003:**
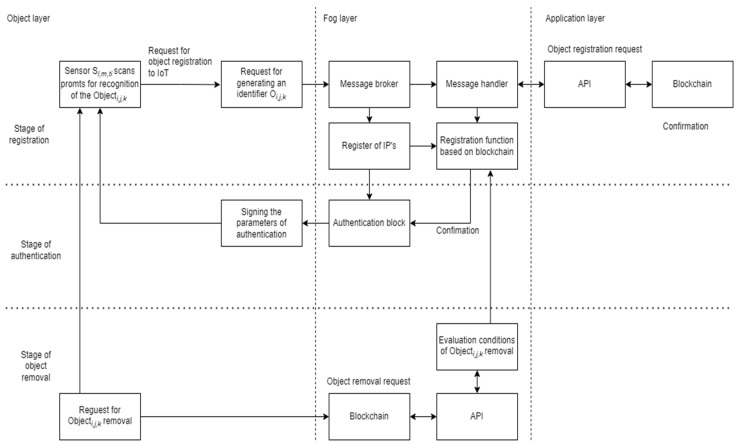
Integration of stages for registration, authentication, and removal of moving transport objects in the system based on the IoT technological platform.

**Figure 4 sensors-23-02831-f004:**
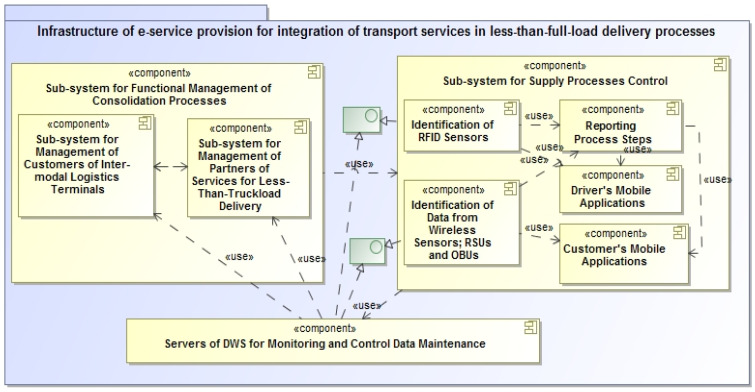
A conceptual model for assessment of transport services in the e-STMS for less-than-truckload delivery process management.

**Figure 5 sensors-23-02831-f005:**
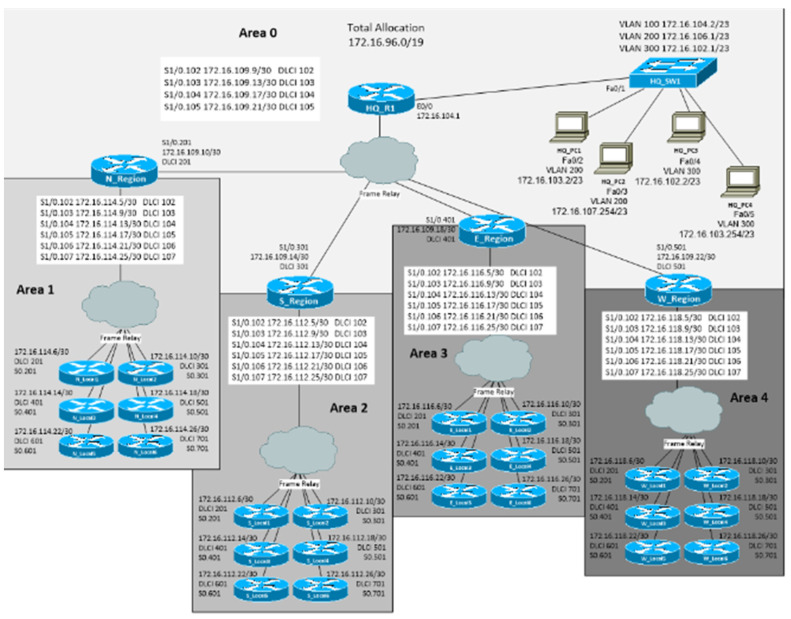
Illustration of NetSIM Lab experimental environment for connection of routers for multi-area access enabling the areas of cloud computing (Source: [137,138]).

**Table 1 sensors-23-02831-t001:** Different perspectives by different analyses of container port operations.

Transportation Content of the Analysis	Sources
Quay planning	[40,41,42,43]
Hours of operation of quay cranes	[44,45,46,47]
Planning and sequence of loading	[48,49]
Storage activities	[50,51]
Distribution and dispatch of cranes and conveyors	[52,53,54]

**Table 2 sensors-23-02831-t002:** The layers of IoT applicable for monitoring intermodal transportation objects.

Different Layers of IoT and Connected Objects	Vulnerability Problems, Which Are Needed to Solve	Proposed Solutions and Measures for Safety Work
Application layer	Data access problems with solving the safety connectivity of nodes and supporting the interoperability of participating data warehouses	- Right support of connectivity between separate, far away nodes of sensors by integrating into the whole infrastructure based on WSNs; - Supporting means for working safety communication channels; - Enabling means of interoperable work of distributed data warehouses and enabling them to control the unification of primary keys and identification functions; - Development of meta-models for support of repositories work and structures of DWs; - Develop means of accessing essential management; - Create the interconnection means to distribute different types DWs
	Security issues related to authentication of the moving objects, rights to access to the DWs, and development of agent’s use cases with rules for access/restriction and allowance to infrastructure	- Development of the distribution algorithms; - Following instructions and developing the set of unsafety detection algorithms; - Applying the cryptography methods; - Development of the authentication means - Support restricted access control;
Network layer	Compatibility issues; Heterogeneity of communication channels; Different availability of clusters and restriction access security problems; Endemic network protocol problems; Privacy disclosure, etc.	- Implementation of compatibility and application of appropriate risers; - Development of secure routing protocols; - Realization of the interconnections of availability access to heterogeneity channels and communication protocols
Perceiving layer	Node capture; Possibilities of unsafe activities and fake nodes Detection problems; Side channel and replay attacks; Mass node authentication problem	- Development of the proper physical design; - Development of means for authentication and access control; - Design concrete algorithms for significant data discovery;

**Table 3 sensors-23-02831-t003:** Analysis of tasks of reporting for management of logistic and transport processes.

Functional Area	Solutions	Solutions Provided by Authors
Security of WNS communication components	Security of cameras, e-seal, gate entrance RFID, cargo location tracking RFID	[131]
Connectivity of transport modes	IoT-based intelligent transportation system (IoT-ITS)	[132]
Availability of services in terminal	Sensors to support the IoT for monitoring of terminal overload	[133]
Energy saving	Green IoT	[134]
Monitoring of cargo storage conditions	Temperature monitoring and alarming RFID devices	[135,136]

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
