# Peer review of "Development of E-Service Provision System Architecture Based on IoT and WSNs for Monitoring and Management of Freight Intermodal Transportation"

_sensors, 2023, doi:10.3390/s23052831_

Round 1

Reviewer 1 Report

dear authors:

the paper is very good, however, I suggest making the following comments 

1- the abstract is very long. reduce it.

2- the result needs more discussion.

Author Response

Reviewer1

Remarks

The paper is very good, however, I suggest making the following comments 

1- the abstract is very long. reduce it.

Authors: The abstract has been updated

Reviewer 1 Question 2- the result needs more discussion. 

Authors: The Methodology for development the system and architecture of the e-service provision system for monitoring and management of cargo intermodal transportation processes is improved.  The Section 7 for illustration of experimental research is extended and added to the Article. The Discussion section has been updated.

Reviewer 2 Report

The objective of the article is to present a solution based on IoT and RFID for logistics. Below are some observations: Improve English. E.g.: “Analysis of processes WHICH are related with …”, and others. Which are the problems which arise during the process of developing this architecture? Good references All figures have a poor quality even in 200% zoom. This can easily be improved by pasting an improved version. In the end you state architecture and conceptualization, but no such thing is presented throughout the article. The only description is “OBU devices communicate with various nodes by providing non-security services using other mobile technologies (GPRS, GSM, 4G, HSDPA, UMTS, and WiMax)”. Sections 2 and 3 which seem to deal with what you call architecture does not present implementation aspects.

Author Response

Reviewer2

Comments and Suggestions for Authors

Focus – results and conclusions

The objectives of the article are improved and presented in Abstract and Introduction. The solving problems are highlighted in Introduction. For presentation of solutions, based on IoT and RFID for logistics and intermodal transportation, are presented by providing more detailed architecture of the e-service provision system based on IoT and WSNs, by paying more attention for scanning of moving objects at wireless sensors and safety including in IoT platform.

Below are some observations:

Improve English. E.g.: “Analysis of processes WHICH are related with …”, and others.

Authors: The English language has been revised.

Which are the problems which arise during the process of developing this architecture?

Authors: The authors added a description of the experimental environment for the connection of routers for multi-area access enabling areas of cloud computing.

Good references

All figures have a poor quality even in 200% zoom. This can easily be improved by pasting an improved version.

Authors: The authors will improve the quality of Figures

In the end you state architecture and conceptualization, but no such thing is presented throughout the article. The only description is “OBU devices communicate with various nodes by providing non-security services using other mobile technologies (GPRS, GSM, 4G, HSDPA, UMTS, and WiMax)”.

Authors: The authors added a citation to previous papers where the such description was presented.

Sections 2 and 3 which seem to deal with what you call architecture does not present implementation aspects.

Authors: The authors added the descriptions. The aspects of  architecture are described. The Methodology for development the system and architecture of the e-service provision system for monitoring and management of cargo intermodal transportation processes is improved.  The Section 7 for illustration of experimental research is extended and added to the Article. The Discussion section has been updated.

Reviewer 3 Report

This paper described an architecture of e-service provision system 

Here are some comments:

1- the abstract is not clear. The contribution itself is not clear to me.  You mentioned in the title that you developed E-Service Provision System then you described an architecture and do not provide any kind of evaluation. The abstract is confusing and not concise. You sometimes mentioned that you develop a system and sometimes mentioned (approach/architecture). Be consistent and concise.

2- the structure of the paper and the sections especially methodology  needs improvement. A lot of details are not clear.

3- some figures are not clear especially fig 2. All figures need to be explained in details.

4- how did you evaluate your work?

5- The methodology section is important. you need to provide the layers you have adopted in clear and logical way, use graphical abstract that illustrate your contribution and describe the layers in clear flow. After that you need to provide an evaluation for the proposed approach?

6- improve the English language used in this paper and use clear words to represent your ideas. For instance “ Introduction focuses on actuality and introduce the paper research area” what do you mean by “actuality”?

Author Response

Reviewer3

This paper described an architecture of e-service provision system 

Here are some comments:

  • the abstract is not clear. The abstract is confusing and not concise. Point 2

Authors: The authors revised the abstract.

  • The contribution itself is not clear to me.  You mentioned in the title that you developed E-Service Provision System then you described an architecture and do not provide any kind of evaluation. You sometimes mentioned that you develop a system and sometimes mentioned (approach/architecture). Be consistent and concise.

Authors: The authors corrected the title of the paper by adding the "Architecture" in the title, and improved the description of architecture and implementation of needful components of infrastructure of other needful technology.

2- the structure of the paper and the sections especially methodology  needs improvement. A lot of details are not clear.

Authors: The methodology section is improved.

3- some figures are not clear especially fig 2. All figures need to be explained in details.

Authors: The authors improved the quality and explained the mentioned Figures.

4- how did you evaluate your work?

The experimental research possibilities of implementation of the proposed architecture were described in Section 7 additionally, showing the applying of the NetSIM Lab equipment as network modelling platform. 

5- The methodology section is important. you need to provide the layers you have adopted in clear and logical way, use graphical abstract that illustrate your contribution and describe the layers in clear flow. After that you need to provide an evaluation for the proposed approach?

Authors: The methodology section is updated. The authors added a graphical illustration.

6- improve the English language used in this paper and use clear words to represent your ideas. For instance “ Introduction focuses on actuality and introduce the paper research area” what do you mean by “actuality”?

Authors: The English language is revised.

Round 2

Reviewer 3 Report

I went through this paper. Although it has been improved since the first submission, however, I still have concerns regarding the clarity of the contribution and the writing. The authors have changed the title of the paper, but in many instances, they mentioned (Development/implementation) such as: Figure 1. Main steps of action for the implementation of needful processes for the development of the e-STMCS I think it needs extensive editing and proof-reading before it can be accepted.

Author Response

The authors corrected the article after checking the English spelling errors. The terms used were also clarified, and the abstract was shortened and corrected. The concepts used and their application in separate parts of the article were reviewed and they were unified. The descriptive architecture of the entire e-service provision system has been refined and linked to the methodology.
The title of Figure 1 has been corrected, clarifying the naming of the steps and what they represent.

The attachment of file with Change Tracks shows the improving process of the article. 
